

# North Atlantic deep water formation and AMOC in CMIP5 models

Heuzé Céline[1]

[1]Department of Marine Sciences, University of Gothenburg, Box 115, 405 30 Göteborg, Sweden

*Correspondence to:* Céline Heuzé (celine.heuze@marine.gu.se)

**Abstract.** Deep water formation in climate models is indicative of their ability to simulate future ocean circulation, carbon and heat uptake, and sea level rise. Present-day temperature, salinity, sea ice concentration and ocean transport in the North Atlantic subpolar gyre and Nordic Seas from 23 CMIP5 (Climate Model Intercomparison Project, phase 5) models are compared with observations to assess the biases, causes and consequences of North Atlantic deep convection in models. The majority of
5 models convect too deep, over too large an area, too often, and too far south. Deep convection occurs at the sea ice edge and is most realistic in models with accurate sea ice extent, mostly those using the CICE model. Half of the models convect in response to local cooling or salinification of the surface waters; only a third have a dynamic relationship between freshwater coming from the Arctic and deep convection. The models with the most intense deep convection have the warmest deep waters, due to a redistribution of heat through the water column. For the majority of models, the variability of the Atlantic Meridional
Overturning Circulation (AMOC) is explained by the volumes of deep water produced in the subpolar gyre and Nordic Seas up to 2 years before. In turns, models with the strongest AMOC have the largest heat export to the Arctic. Understanding the dynamical drivers of deep convection and AMOC in models is hence key to realistically forecast Arctic oceanic warming and its consequences on the global ocean circulation, cryosphere and marine life.

## 1 Introduction

Global fully-coupled climate models are a key tool to study current and future climate change, but although they clearly improve from one generation to the next, they still suffer from many biases (Flato et al., 2013). In particular the horizontal resolution of the ocean, around 1° (Table 1), is too coarse for explicitly representing eddies, freshwater plumes and overflows. Yet all these processes are necessary to correctly generate deep water formation (Marshall and Schott, 1999).

Deep water formation occurs around Antarctica and in the North Atlantic (Killworth, 1983). It is vital to ventilate the
20 ocean and for the global ocean circulation, but also for heat and carbon storage (e.g. Sabine et al., 2004; Lozier et al., 2008; Schmittner and Lund, 2015). Moreover in the North Atlantic, deep water formation is tied to the strength of the Atlantic Meridional Overturning Circulation (AMOC, Böning et al., 2006), which transports heat to the Arctic (Spielhagen et al., 2011). This oceanic heat in turns melts the sea ice and Greenland floating glaciers from below (e.g. Polyakov et al., 2010; Straneo and Heimbach, 2013). Hence, the North Atlantic is a crucial area to assess the ability of current generation climate
models to represent deep water formation.



In this manuscript, we compare present-day deep water formation in 23 state-of-the-art global climate models that participated to the Climate Model Intercomparison Project phase 5 (CMIP5, Taylor et al., 2012). We assess their biases in the representation of deep convection in section 3, explore the possible causes of these biases in section 4, notably buoyancy forcings and sea ice, and estimate the consequences of their biases on the AMOC and heat export to the Arctic in section 5. To

the best of our knowledge, similar tests have been done on the previous generation of climate models (CMIP3, de Jong et al., 2009) and in ocean-only simulations (CORE-II, Danabasoglu et al., 2014), but not yet on CMIP5 models. Yet the magnitude of biases in CMIP5 models has to be known in order to properly simulate changes to the Arctic using the current generation of models, but also to evaluate improvements when CMIP6 model simulations become available (Eyring et al., 2016).

## 2  Data and methods

### 2.1  CMIP5 models

The output of 23 CMIP5 models (Taylor et al., 2012), listed in table 1, were used in this study. In the North Atlantic, all models have approximately the same horizontal grid spacing, varying around 1° in both latitude and longitude. The coarsest resolution is 2° (for the CMCC models) and the highest is 0.4° (for MPI-ESM-MR). Most models have a z-level vertical grid with an average of 40 levels (Table 1). Although 4 models were run on a different type of grid (isopycnic, terrain following or hybrid),

their output were submitted on a regular z-level grid.

In this study, 15 models use only three different sea ice components (Table 1): the Los Alamos sea ice model (CICE; Hunke and Lipscomb, 2008), the GFDL sea ice simulator (SIS; Delworth et al., 2006) and the Louvain-la-Neuve sea ice model (LIM; Fichefet and Maqueda, 1997). The other climate models mostly use the sea ice component of their respective ocean models. Although each climate model has a unique configuration, comparing models which share components -as we do in section 4.2-

can indicate what causes a misrepresentation.

We are interested in the mean, present state of the ocean and hence use twenty years of monthly historical run, from January 1986 to the end of the historical run in December 2005. The monthly pre-industrial control run was used to remove possible model drift. We also use the control run from 1986 to 2100 to study lagged correlations in a subset of 14 models for which such long runs were available (indicated with a star in Table 1) in section 5.2. Only one ensemble member per model was used,

r1i1p1, for it was the only one common to all the models at the date of download (July 2016).

### 2.2  Observational-based products

Three observational-based analysis products are used for assessing the models' representation of the present-day ocean. They are not the most recent climatologies, but have been chosen as representative of the 1986-2005 period studied here with the climate models. The observed monthly climatology of mixed layer depth (MLD) is that of de Boyer Montégut et al. (2004),

available at http://www.ifremer.fr/cerweb/deboyer/mld/home.php. It was created using a density criterion of 0.03 kg m$^{-3}$ over more than 4 million hydrographic profiles, taken from 1941 to 2002, interpolated onto a regular 2° x 2° horizontal grid.



The temperature and salinity of the observed water column are given by the World Ocean Atlas 2009 (WOA09, Locarnini et al., 2013; Zweng et al., 2013, http://www.nodc.noaa.gov/OC5/WOA09/pr_woa09.html). It includes over 9 million quality-controlled hydrographic profiles. The monthly climatology is limited to the top 1500 m of the ocean, hence the seasonal climatology is used here. It is provided as a regular 1° x 1° x 33 level grid.

Finally, we use the HadISST monthly sea ice concentration measurements (Rayner et al., 2003, http://www.metoffice.gov.uk/hadobs/hadisst/), from January 1986 to December 2005, also provided as a regular 1° x 1° grid. The observed sea ice extent is computed as the sum of the areas of the grid cells with a sea ice concentration larger than 15%. To facilitate comparisons, the model output have been interpolated onto the common HadISST-WOA09 grid.

## 2.3   Methods

Some climate models provide a mixed layer depth output, but not the majority of them. For consistency amongst models and with the observations, we instead compute the monthly MLD for each model using the de Boyer Montégut et al. (2004) method. That is, using the monthly temperature and salinity model output to compute the density $\sigma_\theta$, we define the MLD as the depth where the density exceeds that of the reference level (10 m) by 0.03 kg m$^{-3}$.

Following observations, we consider that there is deep water formation or deep convection if the MLD exceeds 1000 m 15 (e.g. Marshall and Schott, 1999; Våge et al., 2009). We divide the North Atlantic into two study areas where in the real ocean different deep waters form (Killworth, 1983): the Greenland-Iceland-Norwegian (GIN) seas (latitude 66 to 80°N, longitude 20°W to 20°E) and the subpolar gyre (SG, latitude 50 to 65°N, longitude 65 to 20°W, see orange boxes on Fig. 1a). The volume of deep water formed by each model is defined as the product of the grid cell area by the MLD, summed over all the grid cells with a MLD deeper than 1000 m in each of these two regions.

One of the buoyancy forcings whose impact on deep convection we study is the freshwater flux from the Arctic through the two sections closest to SG and GIN, Davis and Fram Straits respectively. Following for instance Aagaard and Carmack (1989) these are computed as:

$$FW = \int_A (1 - S/S_{ref})v\,dA. \tag{1}$$

where $S_{ref} = 34.8$ is a reference salinity, S is the monthly salinity field, v the meridional velocity field, and A the corresponding 25 depth-longitude section. The coordinates considered for Davis Straits are 66°N, 70 to 50°W; for Fram Strait, 80°N and 20° to 15°E (green lines on Fig. 1a). Similarly, the heat flux through Fram Strait was computed as:

$$Q = \int_A \rho_0 c_p \theta v\,dA, \tag{2}$$

where $\rho_0 = 1027$ kg m$^{-3}$ is a reference density of water, $c_p = 3.98$ kJ kg K$^{-1}$ is the specific heat capacity of water, and $\theta$ is the monthly temperature field. The other buoyancy forcings that are studied here are the local heat and salt changes by interaction 30 with the atmosphere. These are defined as the month-to-month difference in heat and salt content respectively from the ocean surface to the MLD.





To assess the consequences of deep water formation, we study the hydrographic properties averaged over the same two depth ranges as de Jong et al. (2009):

- the Labrador Sea Water (LSW) layer, 750 to 1250 m depth;

- and the Northeast Atlantic Deep Water (NEADW) layer, 2000 to 2500 m depth.

We shall refer to water found at these two levels in models as North Atlantic Deep Water (NADW), with no further distinction between LSW and NEADW. We do not attempt to define NADW using temperature, salinity or density criteria as is done in observations (e.g. Weaver et al., 1999), since such criteria are not adapted to models that we expect to feature temperature, salinity or density biases. The monthly AMOC is obtained by integrating the meridional velocity at $30°$N through the Atlantic basin from coast to coast, and then over depth using the bottom of the ocean as the reference level. The AMOC is defined as

the maximum southward transport (Cheng et al., 2013).

## 3   The representation of North Atlantic deep water formation in CMIP5 models

### 3.1   Comparison with observations

Deep convection occurs in the North Atlantic in two main areas: in the subpolar gyre, and in the Greenland-Iceland-Norwegian seas (Fig. 1a). It has been measured to extend deeper than 2000 m (Marshall and Schott, 1999), but it does not occur every

15   year in the real ocean. In fact, over the 1986-2005 period of this study, deep convection occurred in the subpolar gyre only from 1987 to 1994 and in winter 1999/2000 (Yashayaev, 2007; Våge et al., 2009); in the GIN seas, only in winter 1988 (Marshall and Schott, 1999). Hence the climatology made of observations shows relatively shallow mean mixed layers that do not exceed 1000 m (Fig. 1a). Still, some models are clearly convecting too deep, with MLD reaching from the surface to the sea floor: GFDL-CM3, GISS-E2-R, IPSL-CM5A-MR and MPI-ESM-MR in the SG area (Fig. 1l,o,s,w); and GISS-E2-R and

IPSL-CM5A-MR again as well as both MIROC in the GIN area (Fig. 1o,s,t,u).

Most models exhibit very deep 20-year mean mixed layers, over large areas, and convect in both regions nearly every year. In the SG region, the models can be split into three different groups based on the location of the deep convection centre:

- the models that convect mostly in the Labrador Sea, or northern part of SG: CCSM4, CESM1-CAM5, CNRM-CM5, FGOALS-g2, HadGEM2-CC and -ES, MIROC5, and MPI-ESM-LR and -MR (Fig. 1e,f,i,k,p,q,t,v,w)

- the models that convect too far in the south: ACCESS1-0, bcc-csm1-1, CanESM2, CMCC-CM and -CMS, GFDL-ESM2G and M, IPSL-CM5A-LR and -MR, MIROC-ESM-CHEM, and NorESM1-M (Fig. 1b,c,d,g,h,m,n,r,s,u,x)

- the models that convect everywhere: CSIRO-Mk3-6-0, GFDL-CM3 and GISS-E2-R (Fig. 1j,l,o)

In fact, in the SG area, 9 out of 23 models convect at the correct location. The majority of models convect at the wrong location, and in particular too far in the south, which is a common feature in climate models (e.g. Treguier et al., 2005; Jungclaus et al.,



2005). In both SG and GIN, the location of deep MLD seem contrained by the winter sea ice extent (cyan line on Fig. 1); this will be further discussed in section 4.2.

Unlike the real North Atlantic Ocean and its "deep convection seesaw" (Oka et al., 2006), i.e. the alternation between deep convection in SG and in the GIN seas, most models convect in both regions at the same time, every year of the study period. The exceptions are:

- in SG, CanESM2, both CMCCs and CSIRO-Mk3-6-0 convect only 75% of the years (left numbers, Fig. 1d,g,h,j), and CNRM-CM5 less than 50% (Fig. 1i);

- in GIN, CNRM-CM5 again, both HadGEM2s, and IPSL-CM5A-LR convect 75% of the years (right numbers, Fig. 1i,p,q,r), and CMCC-CM and FGOALS-g2 less than 50% (Fig. 1g,k).

No CMIP5 model from this study has a variability similar to that observed in the real North Atlantic during 1986-2005, but two models, CMCC-CM and CNRM-CM5 exhibit more variability, in both seas, than the other models.

A full assessment of the impact of resolution and model code changes is not possible with the limited data used here. In fact, this is the motivation for the CORE-II (Danabasoglu et al., 2014) and upcoming OMIP (Griffies et al., 2016) exercises. It is nonetheless interesting to note how differently models which share component behave. For example:

- CMCC-CM and CMCC-CMS differ only in the configuration of the atmospheric code, yet -CMS has a far more intense deep convection region in the GIN seas (Fig. 1g,h). HadGEM2-CC and -ES also differ only slightly in their atmospheric code (ES includes tropospheric chemistry), yet their deep convection behaviours are not obviously different (Fig. 1p,q).

- IPSL-CM5-LR and -MR differ in the resolution of their common atmospheric component (MR is the highest), and the mean MLD is deeper, over a larger area in -MR (Fig. 1r,s). In the meantime, although CCSM4 and CESM1-CAM5 also have different atmosphere models but the same ocean code, their deep convection behaviours in both seas are equivalent (Fig. 1e,f).

- GFDL-ESM2M is more similar to -ESM2G in deep convection characteristics despite their different ocean components than to GFDL-CM3 whose ocean model code is the same as that of GFDL-ESM2M.

In summary, choices of ocean or atmosphere model codes and resolutions cannot be directly linked to specific deep convection behaviours. All models from this study convect too often, too deep and over too large an area when compared to observations. Nine models are relatively realistic though regarding the location of deep convection, and among these nine models four of them, CNRM-CM5, FGOALS-g2, HadGEM2-CC and HadGEM2-ES, also exhibit some temporal variability instead of wrongly convecting each year, and can hence be deemed "the most accurate models".

## 3.2  Has deep convection representation improved since CMIP3?

In a study of eight CMIP3 models, de Jong et al. (2009) found that deep convection was too shallow in the Labrador Sea, and too deep, over too large an area in a region corresponding to the southern part of our SG. Half of the models presented some





variability in the mean maximum MLD, an indication that they did not convect every year in SG. To the best of our knowledge, no study has assessed the performance of CMIP3 models with respects to MLD in the GIN seas, although the occasional map of this region by Carman and McClean (2011) does show a large spread in maximum depth and area of deep convection among the ten models of their study.

CMIP5 models have improved compared to their CMIP3 counterparts since deep convection in the GIN seas is more localised for the majority of them. Nine models of our sample of 24 have realistic MLD in the Labrador Sea, at the correct location, and four of them even have a realistic variability. Most CMIP5 models also convect less deep than the CMIP3 models did; most of the models in the present study convect only to 2000 m on average, whereas most mean CMIP3 MLD in the subpolar gyre extended to the sea floor.

However, some problems remain. The majority of models in our study convect at the wrong location in the subpolar gyre, too far south and/or over too large an area extending south of Iceland. CMIP5 and CMIP3 models alike convect too often, or rather more often than the real ocean did over the same period. And a minority of CMIP5 models has MLD that are far too deep.

Why are some of these biases still present in CMIP5 models? Can they be caused by other biases that have not been improved
and/or specific model components? We investigate these questions now, in section 4.

## 4   Across-model possible causes of deep water formation misrepresentations

### 4.1   Heat and salt

The aim of the present paper is not to determine the dynamics of deep convection in all the individual CMIP5 models, since that would require access to 100-to-1000 year simulations (as was done in the Southern Ocean by Martin et al., 2013, for
example). Instead, we verify whether specific biases in the models are consistently associated with misrepresentations of deep water formation. We concentrate on features that have been highlighted in observations or in other modelling studies as potential triggers for deep convection: freshwater import from the Arctic, local buoyancy forcings and sea ice (Marshall and Schott, 1999).

In this section, we concentrate on the buoyancy biases, both freshwater import from the Arctic and local forcings. No across-
model relationship was found, but rather different behaviours for different models, summarised in table 2. The sign conventions (see methods) are as follows:

- negative for "freshwater from Davis or Fram straits" means that the more freshwater is flowing southward from the Arctic, the less deep convection;

- negative for "Heat Gain" actually means that the stronger the heat loss to the atmosphere (i.e. the more strongly negative
a heat gain value), the deeper the convection;

- positive for "Salt Gain" means that the saltier the surface layer becomes, the deeper the convection.





Nine models out of 23 show a negative correlation between the freshwater coming from the Arctic via Davis Strait and the MLD in the subpolar gyre (Table 2, first and fourth columns). For three of these models, CanESM2, MIROC-ESM-CHEM and MPI-ESM-MR, it is even the only meaningful relationship. In the GIN seas, the negative relationship freshwater from the Arctic - MLD is present in only three models. Note that only one model, ACCESS1-0, has a correlation in both regions.

A mysterious positive correlation, i.e. the stronger the freshwater import the deeper the MLD, is found for six models in SG and six models in GIN, with two models common to both regions (bcc-csm1-1 and GISS-E2-R). Correlation does not mean causation, so it is possible that in these models MLD and freshwater import are linked to a third process, for example the sea ice extent.

The relationship between local heat loss to the atmosphere and MLD is more consistent: 11 models exhibit a negative

correlation in SG, and 10 in the GIN seas (Table 2, second and fifth columns). For these models, as is the case in the real ocean (Killworth, 1983), the stronger the surface cooling, the deeper the MLD. For five models (GFDL-ESM2M, IPSL-CM5A-MR, MIROC5, MPI-ESM-LR and NorESM1-M), the opposite relationship is found: deep convection corresponds to a surface heat gain. It can be that in these models, deep convection is triggered so fast that we see its result, the mixing up of subsurface warm water (Marshall and Schott, 1999), when other models are still in the preconditioning phase. Output at a higher temporal

resolution than the current monthly data would be required to study this question.

Finally, 12 models have deep MLD in association with a salinification of the surface waters in SG, and only three in the GIN seas (positive correlation in the third and sixth columns of Table 2). In fact in the GIN seas, the opposite relationship is encountered the most often, for 12 of the models. CMCC-CMS also has a positive relationship with the freshwater from the Arctic, suggesting that a more-complex-than-thought freshwater cycle in the GIN seas could be linked to deep convection. For

the 11 other models, this negative relationship remains unexplained for monthly output are not high-enough a resolution for such a study.

In summary, in the real North Atlantic, deep convection is mainly controlled by local surface buoyancy forcings: heat loss to the atmosphere in SG (Marshall and Schott, 1999), and haline convection in the GIN seas (Rudels and Quadfasel, 1991). In CMIP5 models, no consistent behaviour was found. CSIRO-Mk3-6-0 is the only model which seems to have the same drivers

of deep convection as the observations, and a clear distinction between the two regions. Half of the models exhibit unexpected relationships, showing that higher resolution output are required to study their dynamics. And five models had no significant correlation; their deep convection hence is probably controlled by something else...

## 4.2 Sea ice

The link between deep convection and sea ice is evident in the GIN seas, where in the real ocean ice crystals have to form in

the surface layer and then rise while saline droplets sink, triggering convection (Rudels and Quadfasel, 1991). The relationship has also been identified in the models from the CORE-II experiments. Danabasoglu et al. (2014) found that models with less sea ice had a salty bias at the surface and hence deeper MLDs.



In the current study, no across-model relationship was found between sea ice extent and deep convection in CMIP5 models. The maximum extent, the seasonal cycle and the variability yielded no significant result. However, we do observe that deep convection follows the winter sea ice edge (cyan lines on Fig. 1), in agreement with Danabasoglu et al. (2014).

In fact, the majority of models that do not convect in the Labrador Sea are ice-covered in this region in winter. Bcc-csm1-1, CMCC-CM, CMCC-CMS, both GFDL-ESM2G and M, IPSL-CM5A-LR and -MR, and MIROC-ESM-CHEM have a sea ice cover between Greenland and North America that extends significantly further south and east than in observations (Fig. 1c,g,h,m,n,r,s,u). Similarly in the GIN seas, the location of deep convection is immediately east of the sea ice. The models which are most ice covered in the GIN seas, notably bcc-csm1-1 and the three GFDL models, have deep convection more in the east and south than the observations (Fig. 1c,l,m,n).

The models with the most accurate representation of deep convection, at least the most accurate location, seem to be the ones with the most accurate winter sea ice extent. In this study, 15 out of the 23 models share only three different sea ice components (Table 1). Seven of them in particular use CICE: ACCESS1-0, CCSM4, CESM1-CAM5, FGOALS-g2, both HadGEM2 models, and NorESM1-M. Although ACCESS1-0 and NorESM1-M convect too far south in the subpolar gyre (Fig. 1b,x), these seven models are amongst the most accurate in this study. Future model intercomparison effort should consider studying the effect of the sea ice model on the ocean. In fact, such is the plan of the upcoming SIMIP (Notz et al., 2016).

The present study does not mean to identify the driving mechanism for deep convection in the North Atlantic in CMIP5 models. In fact, it has proven that such an exercise is not possible with this type of output, and that dedicated modelling exercises should be performed instead. In a last result section, we shall see why they should indeed be performed, i.e. which impacts a misrepresentation of deep convection has on the water column and ocean circulation.

# 5 Why inaccurate North Atlantic deep water formation is a problem

## 5.1 Consequences on the water column

Following de Jong et al. (2009), Figure 2 shows the across-model relationship between mean MLD and density biases at two depth ranges representative of the North Atlantic deep waters in the two regions SG and GIN. We find no consistent significant relationship (Fig. 2). For example, the models with the deepest MLD are not the densest. In SG in both layers (Fig. 2a,b), models with the smallest biases tend to be those with a mean MLD deeper than 2000 m, although bcc-csm1-1, FGOALS-g2 and GISS-E2-R are notable exceptions with biases larger than 0.2 kg m$^{-3}$. bcc-csm1-1 is also an exception in the GIN seas, where similarly most accurate models seem to correspond to deep mixed layers (Fig. 2c,d). It can be noted that the majority of models have relatively accurate NADW densities at both depths: 13 models are within 0.1 kg m$^{-3}$ of the observations in the SG area (Fig. 2a,b), and 17 in the GIN seas (Fig. 2c,d).

As was already the case in CMIP3 models (de Jong et al., 2009), there is no clear relationship between the water column density and deep convection, but there is a relationship with temperature in the subpolar gyre (Fig. 3a,b). At both depth levels, the deeper the mean mixed layer, the warmer the model. The relationship is the strongest below 2000 m (R=0.59, Fig. 3b),




where the temporal spread in the temperature values is also lower. In the GIN seas, no significant relationship could be found between the temperature and the MLD, at either depth (Fig. 3c,d).

In summary, dense water formation is not associated with specific density biases (Fig. 2), and if anything it is linked to deep warm biases (Fig. 3). To explain this seemingly counter-intuitive finding, we assess using Fig. 4 how the temperature (a)

and density (b) is reorganised from month to month through the water column, and show only one model. Each year, deep convection occurs in two times (Fig. 4a):

- first, a warming from the surface, where the warming is the strongest, to approximately 500 m depth;

- then, when the MLD is maximum, a cooling from the surface to a certain depth, and a warming below that depth.

For the events with very deep MLD such as those of 1987 to 1990 and 1993 on Fig. 4, the cooling happens through most of

the ML, whereas during shallower events the cooling is limited to the top 500 m of the water column. In fact, during deep convection, heat is merely reorganised through the water column.

Density does increase during deep convection events (Fig. 4b), but also decreases as deep convection is triggered and in the months before. In agreement with the temperature results, it also decreases during deep convection at the depths levels where temperature increases. In fact, in the subpolar gyre in CMIP5 models, deep convection allows the mixing through the water

column of the comparatively warm and salty pool that sits around 500 m. Hence deep convection is associated with a warming of the deep waters, but like in CMIP3 models this warming is compensated by salinity so that there is no consensus regarding density (de Jong et al., 2009).

## 5.2   Consequences on the AMOC and heat export to the Arctic

In CCSM4, Jahn and Holland (2013) found that less deep convection in the North Atlantic leads to a reduced AMOC. Similarly

in CORE-II experiments, deep mixed layers were associated with large AMOC (Danabasoglu et al., 2014). In the current paper, no across-CMIP5 model relationship was found between the mean winter MLD or volume of deep convection in either region and the AMOC.

This lack of result was actually not that surprising. In the real North Atlantic, the AMOC is the result of the combined effects of both deep convection regions (Yashayaev, 2007). Moreover, there is a lag between deep convection and the subsequent

AMOC strength (Jahn and Holland, 2013). We evaluated such lag on a subset of 14 CMIP5 models for which we could obtain 100-year pre-industrial control runs (Fig. 5). The majority of these models exhibit a significant correlation with the AMOC when deep convection in the gyre lags by 0 to 2 years before (Fig. 5, y-axis). There are three maxima, associated with a lag in the GIN seas of 0 to 1 year, but also 14 years before the AMOC (Fig. 5, x-axis).

To the best of our knowledge, this is the first time that this relationship is evaluated in climate models, and the exact timing

is not known either in observations. The large range of values associated with the GIN seas deep convection can be due to the wrong representation of overflows, caused by the too coarse resolutions of the models (Jungclaus et al., 2013). This issue is known, hence possible solutions such as pipe parameterisations that artificially transport the water undisturbed down an





overflow area have been designed (Danabasoglu et al., 2010) and are expected to lead to better representations of the AMOC in future CMIPs.

So there is a relationship between deep convection and the strength of the AMOC in CMIP5 models. In other models, the stronger the AMOC, the more heat is sent northwards to the Arctic (e.g. Jahn and Holland, 2013; Danabasoglu et al., 2014).
We find the same result in CMIP5 models (Fig. 6). The mean heat flux through Fram Strait is not clearly related to the mean volume of deep convection in the subpolar gyre (Fig. 6a), the GIN seas (Fig. 6b), or the sum of the two (Fig. 6c). But there is a strong robust across-model relationship between the AMOC and the heat flux: the stronger the AMOC, the more heat is exported to the Arctic through Fram Strait (Fig. 6d).

In most CMIP5 models, there is a dynamical relationship between deep water formation and the AMOC. There is also a
relationship between the AMOC and the heat export to the Arctic. So not only the volumes but also the temporal variability of deep convection need to be better represented to correctly model the amount of oceanic heat that enters the Arctic through Fram Strait.

## 6   Conclusions

CMIP5 models have improved their representation of deep convection in the North Atlantic compared to CMIP3 models
(de Jong et al., 2009). Nearly half of them convect at the correct location, and a third of them with some variability -as do the observations. The rest convects too often, too deep, and too far south in the subpolar gyre (Fig. 1). The cause for deep convection bias is model-dependent. The depth is linked to buoyancy forcings for more than half of the models (Table 2); the area and location, to the sea ice extent (Fig. 1). In particular, models with the same sea ice component, CICE, seem to have the most accurate sea ice extent in the subpolar gyre and Greenland-Iceland-Norwegians seas, and the most accurate deep
convection there. We found that deep convection leads to a redistribution of heat through the water column, so that the models with most intense convection are in fact the warmest (Fig. 3); nothing consistent was found regarding density because of salinity mixing. Finally, the stronger the deep convection in CMIP5 models, the stronger their Atlantic Meridional Overturning Circulation two years later (Fig. 5), and in turns the stronger the heat export to the Arctic (Fig. 6).

These results should be taken as they are: correlations, not full dynamical studies. Dedicated experiments, performed on
an ocean at rest, over centuries, would be needed to assess what triggers deep convection in each model, and would probably require output at a higher time resolution than monthly means. Similarly, the relationship between the choice of a sea ice model and the accuracy of deep water formation would need a proper sea ice MIP to be checked. Fortunately, a SIMIP exercise is indeed planned for CMIP6 (Notz et al., 2016). Only then can we accurately assess the heat transport to the Arctic and its future change, and hence predict the demise of Arctic sea ice and Greenland floating glaciers.



## 7 Data availability

The CMIP5 data are available at http://www.ipcc-data.org/sim/gcm_monthly/AR5/Reference-Archive.html. The climatological data for the mixed layer depth are available at http://www.ifremer.fr/cerweb/deboyer/mld/home.php, and those of the ocean temperature and salinity at http://www.nodc.noaa.gov/OC5/WOA09/pr_woa09.html. Sea ice concentration observations are
5  available at http://www.metoffice.gov.uk/hadobs/hadisst/.

*Competing interests.*  The author declares that they have no conflict of interest.

*Acknowledgements.*  We acknowledge the World Climate Research Programme's Working Group on Coupled Modelling, which is responsible for CMIP, and we thank the climate modelling groups (whose models are listed in Table 1 of this paper) for producing and making available their model output. The author would like to thank Matthew Palmer of UK MetOffice for his willingness to share model output
10  (that were eventually not used for this manuscript), and Anna Wåhlin for advice and helpful comments.



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



**Table 1.** List of CMIP5 models (Taylor et al., 2012): modelling groups, model names, ocean resolution in the North Atlantic (longitude / latitude / number of depth levels), type of vertical grid in the ocean (z is geopotential, z* is geopotential with free sea surface, $\sigma$ is terrain following, $\sigma_2$ is isopycnic, and H denotes an hybrid grid), and sea ice component. Stars * indicate the models whose pre-industrial control run is used in section 5.2.

| Modelling group | Model name | resolution (x/y/L) | grid | Sea ice model |
|---|---|---|---|---|
| CSIRO and Bureau of Meteorology, Australia | ACCESS1-0 | 1°/1°/50 | z | CICE v4 |
| Beijing Climate Center, China Meteorological Administration | * bcc-csm1-1 | 1°/1°/40 | z | SIS |
| Canadian Centre for Climate Modelling and Analysis | * CanESM2 | 1.5°/1.5°/40 | z | CanSIM1 |
| National Center for Atmospheric Research | CCSM4 | 1°/0.5°/60 | z | CICE v4 |
| | CESM1-CAM5 | 1°/0.5°/60 | z | CICE v4 |
| Centro Euro-Mediterraneo sui Cambiamenti Climatici | CMCC-CM | 2°/2°/31 | z | LIM2 |
| | CMCC-CMS | 2°/2°/31 | z | LIM2 |
| Centre National de Recherches Météorologiques / Centre Européen de Recherche et Formation Avancée en Calcul Scientifique | * CNRM-CM5A | 0.7°/0.7°/42 | z | GELATO v5 |
| CSIRO and Queensland Climate Change Centre of Excellence | * CSIRO-Mk3-6-0 | 1.8°/0.9°/31 | z | component of Mk3 |
| LASG, Institute of Atmospheric Physics, Chinese Academy of Sciences and CESS, Tsinghua University | FGOALS-g2 | 1°/1°/30 | z* | CICE v4 |
| NOAA Geophysical Fluid Dynamics Laboratory | GFDL-CM3 | 1°/1°/50 | z* | SISp2 |
| | * GFDL-ESM2G | 1°/1°/63 | $\sigma_2$ | SISp2 |
| | * GFDL-ESM2M | 1°/1°/50 | z* | SISp2 |
| NASA Goddard Institute for Space Studies | * GISS-E2-R | 1.25°/1°/32 | z* | Russell sea ice |
| Met Office Hadley Centre | * HadGEM2-CC | 1°/1°/40 | z | based on CICE |
| | * HadGEM2-ES | 1°/1°/40 | z | based on CICE |
| Institut Pierre-Simon Laplace | * IPSL-CM5A-LR | 2°/2°/31 | z | LIM2 |
| | * IPSL-CM5A-MR | 2°/2°/31 | z | LIM2 |
| JAMSTEC Atmosphere and Ocean Research Institute (The University of Tokyo), and National Institute for Environmental Studies | * MIROC5 | 0.5°/0.5°/50 | H $\sigma$-z | component of COCO3.4 |
| | * MIROC-ESM-CHEM | 1.4°/1.4°/44 | H $\sigma$-z | component of COCO3.4 |
| Max-Planck-Institut für Meteorologie | * MPI-ESM-LR | 1.5°/1.5°/40 | z | component of MPI-OM |
| | MPI-ESM-MR | 0.4°/1.5°/40 | z | component of MPI-OM |
| Norwegian Climate Centre | NorESM1-M | 1.125°/1.125°/53 | H $\sigma_2$-z | CICE v4 |



**Figure 1.** North Atlantic, a) climatological mixed layer depth of de Boyer Montégut et al. (2004); b to x) mean 1986-2005 winter MLD in the CMIP5 historical run. Orange boxes on a) show the subpolar gyre (SG) and Greenland-Iceland-Norwegian seas (GIN) regions as defined in this study; green dashed lines, Davis and Fram straits. Yellow dotted line on each panel indicates the 1000 m isobath; cyan and magenta lines, the mean March and September sea ice extent respectively. Left number is the number of years, out of 20, with deep convection in SG; right number, deep convection in GIN.





**Table 2.** For each model, for the subpolar gyre SG (left) and the GIN seas (right), significant correlation (if any) between the time series of the winter mixed layer depth and the Arctic freshwater export via Davis (left) or Fram (right) straits, the local heat exchange with the atmosphere and the local surface salinity change.

| Model | SG | | | GIN | | |
|---|---|---|---|---|---|---|
| | Davis | Heat Gain | Salt Gain | Fram | Heat Gain | Salt Gain |
| ACCESS1-0 | -0.43 | -0.50 | - | -0.44 | -0.41 | -0.40 |
| bcc-csm1-1 | 0.54 | -0.51 | 0.43 | 0.55 | -0.42 | 0.48 |
| CanESM2 | -0.53 | - | -0.43 | - | -0.57 | - |
| CCSM4 | - | -0.84 | - | - | -0.62 | -0.51 |
| CESM1-CAM5 | 0.51 | -0.49 | - | - | -0.41 | -0.40 |
| CMCC-CM | - | - | - | - | - | - |
| CMCC-CMS | - | -0.48 | 0.41 | 0.86 | - | -0.45 |
| CNRM-CM5 | -0.85 | - | 0.91 | 0.64 | - | - |
| CSIRO-Mk3-6-0 | - | -0.66 | - | - | - | 0.67 |
| FGOALS-g2 | -0.39 | -0.52 | - | - | - | - |
| GFDL-CM3 | 0.73 | -0.47 | 0.52 | -0.54 | -0.50 | -0.45 |
| GFDL-ESM2G | - | -0.50 | 0.43 | 0.68 | -0.49 | -0.40 |
| GFDL-ESM2M | - | 0.71 | 0.61 | -0.55 | -0.75 | -0.74 |
| GISS-E2-R | 0.41 | -0.51 | 0.42 | 0.55 | 0.39 | -0.45 |
| HadGEM2-CC | -0.55 | -0.48 | - | - | - | - |
| HadGEM2-ES | - | - | 0.41 | - | -0.72 | -0.39 |
| IPSL-CM5A-LR | - | - | - | - | -0.55 | -0.55 |
| IPSL-CM5A-MR | 0.46 | 0.48 | 0.70 | - | 0.49 | -0.48 |
| MIROC5 | -0.64 | 0.61 | 0.53 | 0.51 | - | - |
| MIROC-ESM-CHEM | -0.49 | - | 0.60 | - | - | - |
| MPI-ESM-LR | -0.40 | 0.43 | 0.55 | - | - | 0.62 |
| MPI-ESM-MR | -0.48 | - | -0.42 | 0.48 | - | - |
| NorESM1-M | 0.45 | 0.69 | - | - | 0.49 | -0.51 |







**Figure 2.** Across-model relationship between the 20-year mean density bias at the two depth levels representative of NADW (columns) and the 20-year mean winter MLD, in the subpolar gyre SG (top) and the GIN seas (bottom).





**Figure 3.** Across-model relationship between the 20-year mean temperature bias at the two depth levels representative of NADW (columns) and the 20-year mean winter MLD, in the subpolar gyre SG (top) and the GIN seas (bottom).



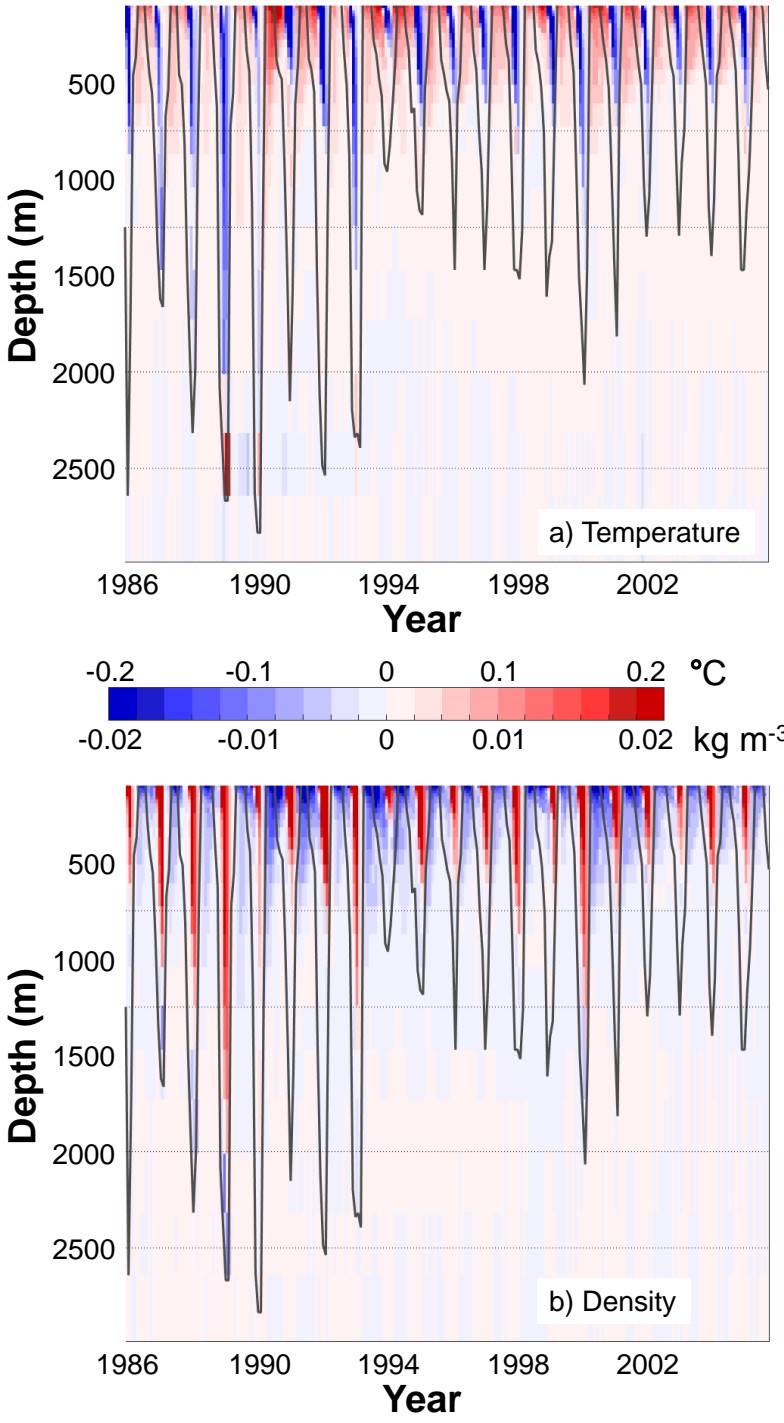

**Figure 4.** For only one model, CanESM2, Hovmöller diagram showing the difference from one month to the next of the temperature (top) and density (bottom) profiles with depth in the subpolar gyre. Dark grey line represents the SG mixed layer depth. Black dotted vertical lines highlight the depth levels representative of NADW: 750, 1250, 2000 and 2500 m.



**Figure 5.** Number of models out of the 14 model-subset where a significant correlation was found between the AMOC and the sum of the volumes of deep convection SG + GIN, for different lags of SG (vertical) and GIN (horizontal); deep convection before AMOC.







**Figure 6.** Across-model relationship between the 20-year mean heat export to the Arctic through Fram Strait and the 20-year mean a) volume of deep convection in SG; b) volume of deep convection in GIN; c) total volume of deep convection SG+GIN; d) AMOC.