# Peer review of "North Atlantic deep water formation and AMOC in CMIP5 models"

_Ocean Science, 2017_

## Referee Comment (RC1) · P. R. Gent (Referee) · 14 Mar 2017

This manuscript examines North Atlantic deep water formation and its association with AMOC in 23 CMIP5 climate models. Much variability is found in the location, timing and strength of deep water formation. For example, only 9 out of the 23 models show deep water formation in the Labrador Sea, and not out in the Subpolar Gyre. Even so, the conclusion is that the CMIP5 models have improved compared to the CMIP3 models.

Figure 2 compares the mean mixed-layer depth versus density bias at two depths in the Subpolar Gyre and the GIN Seas, and no obvious relation is found. I think the MLD would be more related to the vertical density gradient, rather than the density itself. Too deep a MLD is probably related to too small a vertical density gradient in the deep

ocean below about 1500m. Models that convect to the ocean bottom probably have very weak density gradients throughout the whole column. The mean temperature bias at the same locations is shown in Fig 3. I would like to see the mean salinity bias as well, because salinity is more important is setting the density when the temperature is this low.

I would also like to see finer temporal resolution in Fig 4, as I'm unsure whether the warming is causing the MLD errors, or whether the MLD errors are causing the warming. The lag of 2 years between Subpolar Gyre convection and AMOC strength in Fig 5, and the fact that the Fram Strait heat flux is proportional to AMOC in Fig 6 have been documented before; please add some references.

Probably the most useful comment for modelling groups is that they need to get the winter sea ice extent correct in order to get deep water formation in the right location. Are there any other helpful insights that the author can make to help the modelling groups?

Minor Comments: Page 5, line 1; constrained.

Page 9, line 27: says 3 maxima, but only 2 lags are given on line 28.

---

## Referee Comment (RC2) · Anonymous Referee #2 · 15 Mar 2017

General comments The manuscript investigates the CMIP5 model suite on consistency with observations. This is a follow up on an earlier study by de Jong et al. (2009), who investigated the hydrography in the CMIP3 models. While the big discrepancies found by this earlier study made some observational oceanographers very cautious about climate models, it was generally not picked up by the climate community. It is important to see whether the newer generation models is doing a better job, especially because these model are used more and more to explain observed variability on (interannual) time scales for which the models were not intended.

The correlation between deep water formation and sea ice found by the author seems straightforward. It is clear the sea ice extends to far over the Labrador Sea in several of the models. It would be interesting to know why this is the case, although apparently the ice model is one factor. The (sign of the) correlation deep convection and heat

fluxes in some models is confusing, as in the observations there is no doubt about what this sign should be. Even though it is not in the scope of this study to find out why this is, it should be a warning to modelers.

I do have two remarks about possible causes that should be within the scope of this study. Firstly, the difference in MLD may be due to differences in stratification. Even though their offset in density is small (Section 5.2) their stratification may be off enough to cause significant differences in convection.

Secondly, in the real ocean the stratification is set by eddy exchange between the cold interior and the warm boundary current. At high latitudes, like the Subpolar Gyre, these eddies are not resolved by the climate models. Differences in eddy parameterization may therefore affect the MLD. Some of the CMIP5 models include the newer Fox-Kemper (2008) parametrization that is supposed to address this issue, other do not. This aspect of the models deserves to be investigated and it would be good if Table 1 is expended with a column including information on parameterization.

Minor comments 3.2 Line 30. The study by de Jong seems a bit misquoted here. They did investigate the convection in the Labrador Sea, which was too shallow, but did not investigate where else convection occurred. They cited other studies that suggested this. Please correct.

4.1 Line 27. "..." Best to either replace with one dot or write out explicitly what the author means.

Figure 1. The contour of the ice edge is hard to see in several of the panels. It would be good to make the color of this contour a couple of shades darker. Also, some lines appear to be broken (example panel r), which makes them very hard to see as well. Potentially they'd be clearer if the fonts of the model names were made slightly smaller and the actual plots bigger.

Table 2. It would be good to restate the sign conventions in the table caption.

[Figure]

Figures 2, 3 & 6. Please add some information about the grey lines in the figure captions.

[Figure]

---

## Author Comment (AC1) · 6 May 2017

**This manuscript examines North Atlantic deep water formation and its association with AMOC in 23 CMIP5 climate models. Much variability is found in the location, timing and strength of deep water formation. For example, only 9 out of the 23 models show deep water formation in the Labrador Sea, and not out in the Subpolar Gyre. Even so, the conclusion is that the CMIP5 models have improved compared to the CMIP3 models.**

The author thanks Peter R. Gent for agreeing to review this manuscript. A response to each of his comments is provided below with the following structure:

- The reviewer's comment is repeated, with **bold** font;

[Figure]

- The author's response is given in plain font;

- The corresponding changes to the manuscript are indicated in *italics*.

The role of the reviewers has been acknowledged at the end of the manuscript: *"The author would like to thank [. . .], as well as P.R. Gent and an anonymous reviewer, whose suggestions notably improved the quality of this manuscript."*

**Figure 2 compares the mean mixed-layer depth versus density bias at two depths in the Subpolar Gyre and the GIN Seas, and no obvious relation is found. I think the MLD would be more related to the vertical density gradient, rather than the density itself. Too deep a MLD is probably related to too small a vertical density gradient in the deep ocean below about 1500m. Models that convect to the ocean bottom probably have very weak density gradients throughout the whole column.**

The author agrees with the reviewer. In fact, the stratification was one of the first processes tested for this study, but owing to the lack of across model correlation was not mentioned in the manuscript. This as an error; that has been rectified in the revised version. Since there is no across model relationship, and since the reviewer also suggested to add extra scatter plots for the salinity (see below), the author decided to not add a new figure with the scatter plots of stratification vs MLD to avoid making the manuscript too repetitive. Instead, following a suggestion from Reviewer 2, *the correlation between the MLD and the vertical density gradient for each model has been added to table 2*. These results are discussed in a new paragraph added to section 4.1:

*"Fourteen models out of 23 show a significant, logical relationship between the vertical density gradient and deep convection in the subpolar gyre, and thirteen in the GIN seas, but only nine in both regions (Table 2, grey cells). This relationship MLD - stratification does not correlate with the model MLD biases. For example, CCSM4 and CESM1-CAM have similar deep convection depth and area in the subpolar gyre (Fig. 1e and f), but only CCSM4 has a significant correlation between MLD and vertical*

*density gradient.”*

**The mean temperature bias at the same locations is shown in Fig 3. I would like to see the mean salinity bias as well, because salinity is more important is setting the density when the temperature is this low.**

The reviewer is right that salinity is most important to set the density. *Figures 2 and 3 have been modified as follows in order to show the salinity biases*:

- Fig. 2 now shows the density, temperature and salinity biases in the subpolar gyre;

- Fig. 3 is the same as Fig. 2 but for the GIN seas.

*Section 5.1 has been rewritten* accordingly (discussing SG first, for all parameters, and then GIN), and the following comments on the salinity biases have also been added:

*“As was to be expected in a region where salinity dominates the density signal, the salinity biases resemble the density biases (Fig. 2e,f). As such, no relationship is found between the salinity and the MLD in the subpolar gyre.*

[. . .]

*A similar result is found for salinity biases in the GIN seas (Fig. 3e,f). There no significant across model relationship between MLD and salinity biases, and the most extreme biases are encountered for similar MLD. FGOALS-g2, the saltiest, and CNRM-CM5, the freshest, both have a mean MLD of approximately 1000 m (Fig. 3e).*

[. . .]

*In fact, most models have a warm and salty bias in both seas (Figs. 2 and 3), but those compensate in density.”*

**I would also like to see finer temporal resolution in Fig 4, as I'm unsure whether the warming is causing the MLD errors, or whether the MLD errors are causing the warming.**

[Figure]

The author agrees with the reviewer. As the manuscript already states, a higher temporal resolution is needed to perform a proper causality study. Unfortunately, such output have not been archived for the ocean realm in CMIP5 models, hence monthly means are the best that can be used. As a result, no change has been made to the manuscript to address this point.

**The lag of 2 years between Subpolar Gyre convection and AMOC strength in Fig 5, and the fact that the Fram Strait heat flux is proportional to AMOC in Fig 6 have been documented before; please add some references.**

*References to Delworth et al. (1993), Menary et al. (2012), Lohmann et al. (2014) and Ba et al. (2014) have been added to section 5.2.*

**Probably the most useful comment for modelling groups is that they need to get the winter sea ice extent correct in order to get deep water formation in the right location. Are there any other helpful insights that the author can make to help the modelling groups?**

Reviewer 2 advanced the hypothesis that the different vertical mixing parameterisations used by the different models may be linked to the representation of polar mixed layers. Of particular interest was whether the new Fox-Kemper (2011) parameterisation would be associated with better mixed layers. No such result was found, but the following comments have been added to the manuscript:

*"Moreover, there is no apparent relationship between stratification, MLD, and the vertical mixing parameterisation. In particular, the parameterisation designed by Fox-Kemper et al. (2011) to improve the mixed layer representation (present in the models marked with a black bullet point, Table 2) do not perform consistently better than those with other parameterisations. This lack of relationship between MLD biases and vertical mixing parameterisation was already found by Huang et al. (2014) for the summer MLD."*

[Figure]

**Minor Comments: Page 5, line 1; constrained.**

The typo has been corrected.

**Page 9, line 27: says 3 maxima, but only 2 lags are given on line 28.**

The typo has been corrected.

---

## Author Comment (AC2) · 6 May 2017

**General comments**

The manuscript investigates the CMIP5 model suite on consistency with observations. This is a follow up on an earlier study by de Jong et al. (2009), who investigated the hydrography in the CMIP3 models. While the big discrepancies found by this earlier study made some observational oceanographers very cautious about climate models, it was generally not picked up by the climate community. It is important to see whether the newer generation models is doing a better job, especially because these model are used more and more to explain observed variability on (interannual) time scales for which the models were not intended.

The author is really grateful to the reviewer for their extremely encouraging comment!

A response to each of their comments is provided below with the following structure:

- The reviewer's comment is repeated, with **bold** font;

- The author's response is given in plain font;

- The corresponding changes to the manuscript are indicated in *italics*.

The role of the reviewers has been acknowledged at the end of the manuscript: *"The author would like to thank [...], as well as P.R. Gent and an anonymous reviewer, whose suggestions notably improved the quality of this manuscript."*

**The correlation between deep water formation and sea ice found by the author seems straightforward. It is clear the sea ice extends too far over the Labrador Sea in several of the models. It would be interesting to know why this is the case, although apparently the ice model is one factor.**

The author had the opportunity to present her manuscript and discuss this question last week at EGU. The misrepresentation of the Arctic sea ice is a known fact, and fixing it an active area of research. The author and collaborators have designed a new study to investigate one potential cause of this misrepresentation linked to the ocean model component, and hopes to have results to report in a future manuscript.

**The (sign of the) correlation deep convection and heat fluxes in some models is confusing, as in the observations there is no doubt about what this sign should be. Even though it is not in the scope of this study to find out why this is, it should be a warning to modelers.**

The reviewer is right, and a sentence has been added to that effect in the conclusions to highlight this result:

*"Surprisingly, some models exhibited counter-intuitive, unobserved relationships between freshwater fluxes, local buoyancy forcings and mixed layer depth (Table 2); ded-*
*icated studies should be performed by the modelling community to assess the causes of such spurious relationships."*

Note also that to emphasise the results of Table 2 and make the table easier to read, *colours have been added* when the correlation was of the "correct" sign, and left white otherwise (see also one of the minor comments below).

**I do have two remarks about possible causes that should be within the scope of this study. Firstly, the difference in MLD may be due to differences in stratification. Even though their offset in density is small (Section 5.2) their stratification may be off enough to cause significant differences in convection.**

This comment was also made by Reviewer 1. *The correlation between the MLD and the stratification for each model has been added to table 2.* These results are discussed in a new paragraph added to section 4.1:

*"Fourteen models out of 23 show a significant, logical relationship between the vertical density gradient and deep convection in the subpolar gyre, and thirteen in the GIN seas, but only nine in both regions (Table 2, grey cells). This relationship MLD - stratification does not correlate with the model MLD biases. For example, CCSM4 and CESM1-CAM have similar deep convection depth and area in the subpolar gyre (Fig. 1e and f), but only CCSM4 has a significant correlation between MLD and vertical density gradient."*

**Secondly, in the real ocean the stratification is set by eddy exchange between the cold interior and the warm boundary current. At high latitudes, like the Subpolar Gyre, these eddies are not resolved by the climate models. Differences in eddy parameterization may therefore affect the MLD. Some of the CMIP5 models include the newer Fox-Kemper (2008) parametrization that is supposed to address this issue, other do not. This aspect of the models deserves to be investigated and it would be good if Table 1 is expended with a column including information on parameterization.**
The author thanks the reviewer for this suggestion. When investigating the vertical mixing parameterisation of each of the 23 models of this study in order to address the Reviewer's comment, the author discovered that such a study had already been performed by Huang et al. (2014). Rather than paraphrasing their findings, the author decided here to concentrate only on the models with the Fox-Kemper parameterisation, as suggested by the Reviewer. *These models are marked with a black bullet point in Table 2*, and the following text has been added:

*"Moreover, there is no apparent relationship between stratification, MLD, and the vertical mixing parameterisation. In particular, the parameterisation designed by Fox-Kemper et al. (2011) to improve the mixed layer representation (present in the models marked with a black bullet point, Table 2) do not perform consistently better than those with other parameterisations. This lack of relationship between MLD biases and vertical mixing parameterisation was already found by Huang et al. (2014) for the summer MLD."*

**Minor comments**

**3.2 Line 30. The study by de Jong seems a bit misquoted here. They did investigate the convection in the Labrador Sea, which was too shallow, but did not investigate where else convection occurred. They cited other studies that suggested this. Please correct.**

*This sentence now refers to Drijfhout et al. (2008)*, which was the study cited by de Jong et al. (2009) with respects to modelled deep convection in the subpolar gyre, and which had been misquoted by the author.

**4.1 Line 27. "..." Best to either replace with one dot or write out explicitly what the author means.**

*This has been rewritten and now explicitly states: "We now check if this could be the sea ice."*

**Figure 1. The contour of the ice edge is hard to see in several of the panels. It would be good to make the color of this contour a couple of shades darker. Also, some lines appear to be broken (example panel r), which makes them very hard to see as well. Potentially they'd be clearer if the fonts of the model names were made slightly smaller and the actual plots bigger.**

Following the Reviewer's suggestion, *the following changes have been made to Fig. 1*:

- the winter sea ice contours, previously cyan, are now darker;

- the model names fonts are smaller;

- on each subpanels, the maps are larger.

**Table 2. It would be good to restate the sign conventions in the table caption.**

*The sign convention is now restated, and colours have been added to the cells to make the table easier to read.*

**Figures 2, 3  6. Please add some information about the grey lines in the figure captions.**

The grey lines are black lines that have changed colour for a mysterious reason as the image was saved by Matlab. *They have now been recoloured so that all the lines are black.*